

# Gas-Chromatography using ice coated fused silica-columns: Study of adsorption of sulfur dioxide on water-ice

Stefan Langenberg[1,*] and Ulrich Schurath[2]

[1]Institut für Physikalische und Theoretische Chemie, University of Bonn, Bonn, Germany
[*]now at: Klinik und Poliklinik für Hals-Nasen-Ohrenheilkunde/Chirurgie, University of Bonn, Germany
[2]Institut für Umweltphysik, University of Heidelberg, Germany

*Correspondence to:* Stefan Langenberg (langenberg@uni-bonn.de)

**Abstract.** The well-established technique of gas chromatography is used to investigate interactions of sulfur dioxide with a crystalline ice film in a fused silica wide-bore column. Peak shape analysis of $SO_2$ chromatograms measured in the temperature range $205 - 265$ K is applied to extract parameters describing a combination of three processes: (i) physisorption of $SO_2$ at the surface; (ii) dissociative reaction with water; (iii) slow uptake into bulk ice. Process (ii) is described by a dissociative Langmuir isotherm. The pertinent monolayer saturation capacity is found to increase with temperature. The impact of process (iii) on $SO_2$ peak retention time is found to be negligible under our experimental conditions.

By analyzing binary chromatograms of hydrophobic n-hexane and hydrophilic acetone, the premelt surface layer is probed in the temperature range $221 - 263$ K possibly giving rise to irregular adsorption. Both temperature dependencies fit simple van't Hoff equations as expected for process (i), implying that irregular adsorption of acetone is negligible in the probed temperature range. Adsorption enthalpies of $-45 \pm 5$ kJ mol$^{-1}$ and $-23 \pm 2$ kJ mol$^{-1}$ are obtained for acetone and *n*-hexane.

Our study was motivated to assess the vertical displacement of $SO_2$ and acetone in the wake of aircraft by adsorption on ice particles and their subsequent sedimentation. Our results suggest that this transport mechanism is negligible.

## 1 Introduction

Adsorption of $SO_2$ on ice surfaces is of interest in the chemistry and physics of the troposphere and stratosphere. In particular, large ice particles in contrails have been shown by LIDAR soundings to settle out fairly rapidly (Schumann, 1994). This gave rise to speculations that sedimentation of ice particles provides a significant mechanism for the vertical displacement of $SO_2$ and possibly other adsorbing trace gases, particularly in the upper troposphere. An analogous mechanism has been addressed for the absorption and desorption of $SO_2$ in raindrops falling through the plume of a power station (Walcek and Pruppacher, 1983). Therefore, Langenberg (1997) investigated the adsorption of the water-soluble aircraft exhaust ingredients $SO_2$ and acetone on ice over a wide temperature range. The data were interpreted in terms of a simple Langmuir model: $SO_2$ is weakly adsorbed at the normal ice surface and much more strongly at active surface sites. Huthwelker et al. (2006) and Crowley



et al. (2010) found that the results of Langenberg were in disagreement with the work Clegg and Abbatt (2001) and other investigators: in contrast to these studies, Langenberg's analysis implied a classical temperature dependence, i.e. more $SO_2$ being adsorbed at lower temperatures. In addition, the surface coverage derived from the data of Langenberg (1997) is about an order of magnitude lower than implied by the data of Clegg and Abbatt (2001). Therefore, we have reanalyzed the experiments

of Langenberg (1997) using more sophisticated linear and nonlinear regression techniques using R (R Core Team, 2016). The experimental results and their reanalysis are published in the peer-reviewed literature for the first time.

The interaction of $SO_2$ with ice can be separated into fast and much slower components, as summarized by Huthwelker et al. (2006), Abbatt (2003) and Crowley et al. (2010). In the works of Chu et al. (2000) and Clegg and Abbatt (2001) the fast interaction was probed. Both used low pressure flow tubes interfaced with a mass spectrometer to measure $SO_2$. The

former studied vapor-deposited ice coatings, while the latter used ice that was prepared by freezing a liquid water film. The experiments of Chu et al. (2000) were performed under non-equilibrium conditions on a time scale $< 10$ ms, revealing an initial uptake coefficient of $\gamma = 10^{-5}$ for $SO_2$ at 191 K.

Clegg and Abbatt (2001) measured adsorption / desorption bursts of $SO_2$ on ice in the range 213 – 238 K using $SO_2$ partial pressures between $10^{-5} - 10^{-3}$ Pa in helium. They could show that the equilibrium amount of $SO_2$ adsorbed on ice scaled

with the square root of its partial pressure above the surface. They concluded that the square root dependence results from fast dissociative adsorption of $SO_2$ at the ice surface:

$$SO_2 + 2H_2O \rightleftharpoons H_3O^+ + HSO_3^-. \tag{R1}$$

Experiments on orders of magnitude longer time scales utilizing packed ice columns in the temperature range 213 – 270 K (Clapsaddle and Lamb, 1989; Sommerfeld and Lamb, 1986; Conklin et al., 1993; Conklin and Bales, 1993) have revealed that

$SO_2$ is eventually incorporated in ice and partially oxidized to $H_2SO_4$. The uptake rate increases with temperature and with a less than linear dependence on the $SO_2$ partial pressure. The experimental data were reanalyzed and interpreted by Huthwelker et al. (2001) in terms of $SO_2$ diffusing into an internal reservoir. For various reasons the packed column experiments were not suitable to study fast reversible adsorption on ice.

A long-standing issue in studies of gas-ice surface interactions is the possible involvement of a premelt layer (Bartels-Rausch

et al., 2014). It is now well established that an extended quasi-liquid layer exists at the surface of pure water ice at temperatures of a few K below the melting point (Dash et al., 2006).

We employed a chromatographic technique to study adsorption of $SO_2$: ice was deposited as the stationary phase as a thin film of $2 - 8$ $\mu$m thickness in a fused silica wide bore column (diameter $2r = 530$ $\mu$m$\pm10\%$, length 10 m). Adsorption on the ice surface was studied by injecting small amounts of the trace gas under study spiked with a non-adsorbing tracer into the

column. Due to adsorption, the trace gas was retained the *adjusted retention time* $t_n$ relative to the non-adsorbing tracer and therefore two peaks appear at the column outlet, where they were detected by a suitable detector. $t_n$ was obtained from the peak maxima.

The slope of the adsorption isotherm was determined by the *peak maxima method* of gas-chromatography (Huber and Gerritse, 1971): from $t_n$ and the column void time $t_0$, the capacity ratio $k'$ was measured as function of partial pressure $p$ of





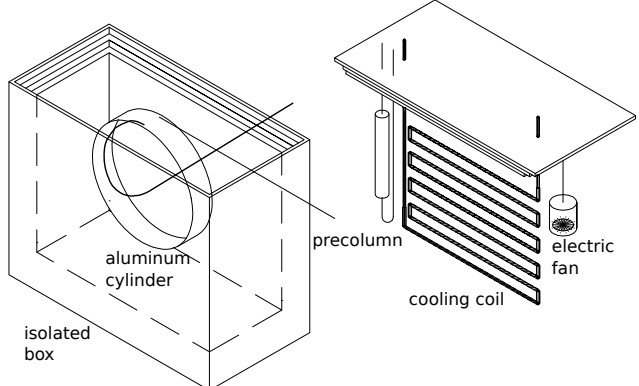

**Figure 1.** Exploded view of the box used for the coating procedure. Inside the box the air is recirculated by a fan (Micronel) to avoid temperature gradients. The temperature is monitored with two Pt-100 sensors mounted at the top and bottom of the box.

the trace gas under investigation. When using a capillary column with cylindrical geometry, $k'$ is related to the slope of the isotherm by

$$k' = \frac{t_n}{t_0} = \left(\frac{\partial n_{ad}}{\partial n_g}\right) = \frac{2RT}{r}\left(\frac{\partial q}{\partial p}\right). \tag{1}$$

$t_0$ is calculated from the column dimensions, the carrier gas mass flow rate and the column head pressure, see e.g. (Langenberg
et al., 1998; Giddings, 1991).

## 2 Experimental

Langenberg and Schurath (1999) described the coating procedure already in detail: a constant volume flow of air ($p = 1 - 2$ bar) was bubbled through a water reservoir at room temperature and admitted into the column maintained at $205 - 220$ K. At the same time the column was slowly drawn through a pinhole into a cold box by a rotating aluminum drum inside the box which
was driven by a stepping motor, see Fig. 1. The box was cooled by a recirculating thermostat (Lauda RLS 6) with a useful temperature range $T \geq 205$ K regulated to $\pm 1$ K. The end of the fused silica column which was fixed to the aluminum drum was connected to a Teflon tube which was led out of the box through the axis of the aluminum cylinder. By slowly drawing the column into the cold box, the water vapor condensed and froze. In this way ice films of 2 - 8 $\mu$m thickness and ca. 8 m length could be prepared.

After finishing the coating procedure, the ice coated column was operated like an ordinary GC column with synthetic air as carrier gas. The flow rate was varied by means of a flow controller (ASM $10 - 100$ sccm) while the column head pressure was monitored with a pressure transducer. The carrier gas from the flow controller was humidified using a precolumn filled with water saturated silica gel that was mounted inside the same cold box.

        Although the carrier gas was humidified to match the vapor pressure over ice inside the cold box, the ice film slowly by
unavoidably is degraded by sublimation due to carrier gas decompression along the length of the column. The rate of ice film





thickness degradation $\dot{h}$ is given by

$$\dot{h} = \frac{V_m \dot{n} p(\text{H}_2\text{O})}{2\pi r l} \left( \frac{1}{p_o} - \frac{1}{p_i} \right), \tag{2}$$

where $p_i$ is the column head pressure, $p_o$ is the pressure at column exit, $l$ is the length of the coated part of the column, $\dot{n}$ the carrier gas mass flow and $V_m = 2 \times 10^{-5} \text{ m}^3 \text{mol}^{-1}$ the molar volume of water-ice. At 265 K, $p_i = 2$ bar, $p_o = 1$ bar, $\dot{n} = 7.85 \times 10^{-5} \text{ mol s}^{-1}$ ($\approx 100$ sccm) is $\dot{h} = 0.7 \mu\text{m h}^{-1}$. Vapor pressure $p(\text{H}_2\text{O})$ over ice is taken from Wexler (1977).

Some experiments were carried out using aged ice-films: after column preparation, the carrier gas was interrupted and the column was kept overnight at 265 K.

Void fused silica columns (SGE) are commercially available either with a pristine silica surface or with a methylsilyl-deactivated surface. Pristine silica surfaces are less hydrophobic than methylsilyl-deactivated silica surfaces. Wetting of the two column types was investigated by contact angle measurements using the capillary rise method (Bartle et al., 1981; Ogden and McNair, 1986). For non wettable surfaces, the contact angle is $\vartheta > 90°$ and for wettable surfaces is $\vartheta < 90°$. For the untreated column $\vartheta = 63 \pm 1°$ and for the methylsilyl-deactivated column $\vartheta = 111 \pm 1°$ was found. To remove ionic impurities, the columns were rinsed with Milli-Q water prior to each coating procedure. The experiments were performed using methylsilyl-deactivated columns unless otherwise indicated. After each coating procedure, series of concentration dependent chromatograms at 2 or 3 constant temperatures were recorded. The temperature was increased in steps of about 15 K. keeping the column for 1 h at each new temperature to achieve thermal equilibrium. In our studies of $\text{SO}_2$ adsorption the peak partial pressure was in the range $p = 0.001 - 1$ Pa at column exit.

$10 - 500 \mu$l dilute mixtures of the compound under study and an appropriate inert tracer were injected with a gas tight syringe. For mixtures of $\text{SO}_2$ and $\text{SF}_6$ a modified Bendix flame photometric sulfur monitor (Farwell and Barinaga, 1986) was used, where $\text{SF}_6$ was the non adsorbable tracer. Acetone, $n$-hexane and methane were monitored using a Carlo Erba flame ionization detector. Here methane was used as non adsorbable tracer. The detector signal was amplified by a Keithley microvoltmeter and recorded by computer.

Due to experimental constraints, about 1 m of the column inside the box remained uncoated. In order to exclude artifacts from this ice-free section the adsorption of $\text{SO}_2$ in a totally ice free column was also investigated. The obtained $k'$ values ranged from $0.35 - 0.45$ at 219 K and $0.15 - 0.25$ at 234 K. At $T > 248$ K $\text{SO}_2$ and $\text{SF}_6$ could no longer be separated. Thus, interference by 1 m ice free column may be neglected for the lower concentration range of the isotherm.

## 3 Results

### 3.1 Adsorption of SO$_2$

For low concentrations the retention times of $\text{SO}_2$ are strongly dependent on the amount of $\text{SO}_2$ injected. The peaks exhibit strong tailing even at low $p(\text{SO}_2)$, see Fig. 2a.




**Table 1.** Nonlinear fit of the experimental data with humidified carrier gas and $h > 4\,\mu$m to different models. The binary variable $\nu$ denotes if the column was aged ($\nu = 1$) or not ($\nu = 0$).

| Model | Model Parameters | $P$ | Residual standard error |
|---|---|---|---|
| Temkin + Henry | $\zeta = (a_1 + a_2 T)(1 + a_3 \nu)$ | | 0.1977 |
| Eq. (7) | $a_1 = -(9 \pm 1) \times 10^{-9}\ \mathrm{mol\,m^{-2}}$ | $< 0.01$ | |
| | $a_2 = (4.8 \pm 0.5) \times 10^{-11}\ \mathrm{mol\,m^{-2}K^{-1}}$ | $< 0.01$ | |
| | $a_3 = 0.6 \pm 0.2$ | $< 0.01$ | |
| | $K_H = (1.11 \pm 0.08) \times 10^{-8}\ \mathrm{mol\,m^{-2}Pa^{-1}}$ | $< 0.01$ | |
| Langmuir + Henry | $K_L = a_1 \exp(a_2/T)$ | | 0.2066 |
| Eq. (10) | $q_S = a_3(1 + a_4 \nu)$ | | |
| | $a_1 = (9 \pm 13) \times 10^{-16}\ \mathrm{mol\,m^{-2}Pa^{-1}}$ | 0.49 | |
| | $a_2 = (5 \pm 0.4) \times 10^{3}\ \mathrm{K}$ | $< 0.01$ | |
| | $a_3 = (1.10 \pm 0.08) \times 10^{-8}\ \mathrm{mol\,m^{-2}}$ | $< 0.01$ | |
| | $a_4 = 0.9 \pm 0.2$ | $< 0.01$ | |
| | $K_H = (1.60 \pm 0.08) \times 10^{-8}\ \mathrm{mol\,m^{-2}Pa^{-1}}$ | $< 0.01$ | |
| Langmuir (Dissociation) + Henry | $q_S = a_1 \exp(a_2/T)(1 + a_3 \nu)$ | | 0.1938 |
| Eq. (16) | $a_1 = (5 \pm 3) \times 10^{-6}\ \mathrm{mol\,m^{-2}}$ | 0.02 | |
| | $a_2 = -(1.3 \pm 0.1) \times 10^{3}\ \mathrm{K}$ | $< 0.01$ | |
| | $a_3 = 0.6 \pm 0.2$ | $< 0.01$ | |
| | $K_{I1}K_H = (7 \pm 5) \times 10^{-14}\ \mathrm{mol^2\,m^{-4}\,Pa^{-1}}$ | 0.1 | |
| | $K_H = (1.35 \pm 0.01) \times 10^{-8}\ \mathrm{mol\,m^{-2}Pa^{-1}}$ | $< 0.01$ | |

For further analysis, the *peak maximum method* (Huber and Gerritse, 1971) is applied to determine $(\partial q/\partial p)$ as function of $p$: it is assumed that Eq. (1) applies to the peak maximum, an assumption which is verified afterwards by peak shape calculation, see discussion.

The most simplistic model for adsorption is Henry's adsorption isotherm: the surface concentration $q$ as function of the gas partial pressure $p$ is given by

$$q = K_H p. \tag{3}$$

After applying Eq. (1) we arrive at

$$k' = \frac{2RT}{r} K_H. \tag{4}$$

If adsorption of a compound is properly described by Henry's adsorption isotherm, the adjusted retention times should be independent of $p(\mathrm{SO_2})$ and thus, of the amount of adsorbing gas injected in our experiments. This is obviously only the case for high $p(\mathrm{SO_2})$. For lower $p(\mathrm{SO_2})$ an additional adsorption mechanism comes into play which yields a $k' \propto p^{-1}$ dependency in the limit $p(\mathrm{SO_2}) \to 0$.





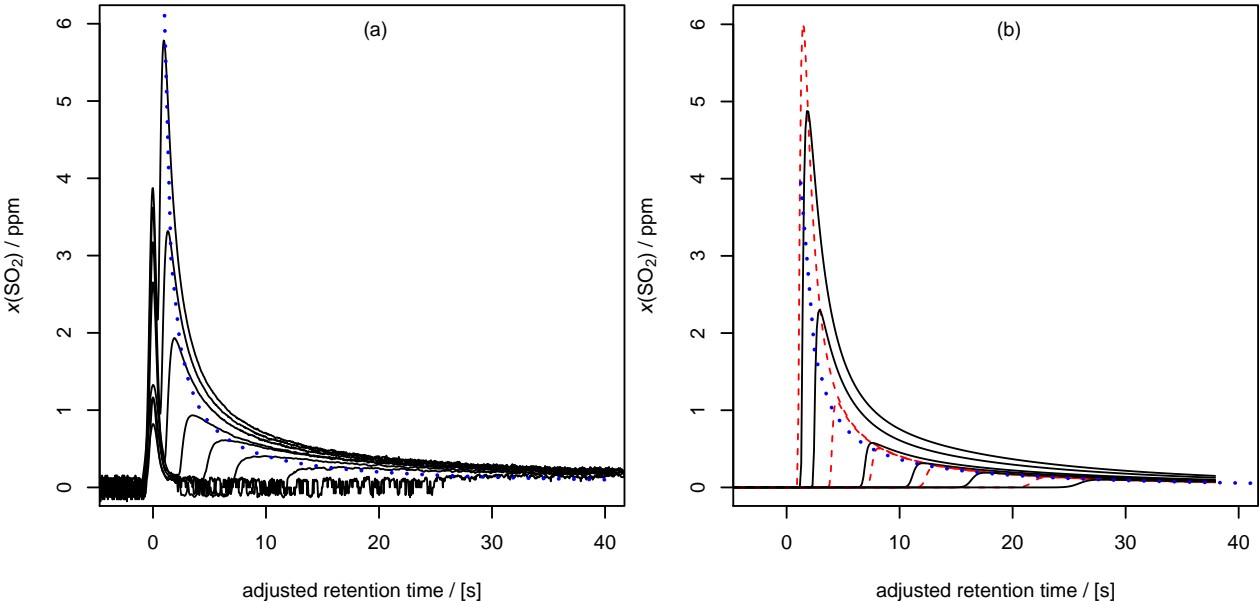

**Figure 2.** (a) Chromatograms from injection of various amounts of a $SF_6/SO_2$ mixture at 247 K and 2 $\mu$m ice film thickness. (b) Simulated $SO_2$ chromatograms using the Temkin isotherm with $\zeta = 4.3 \times 10^{-9}$ mol m$^{-2}$, $K_T = 10^{-6}$ mol m$^{-2}$Pa$^{-1}$, $K_H = 2.8 \times 10^{-9}$ mol m$^{-2}$Pa$^{-1}$ using 100 cells in axial direction. Dashed lines: simulation without absorption/diffusion into the solid phase. Solid lines: simulation including diffusion into the solid phase with $H = 50$, $D_s = 2 \times 10^{-13}$ m$^2$ s$^{-1}$ using 30 cells in radial direction. The dotted lines are calculated with Eq. (1). They represent the locations of the experimental peak maxima.

An isotherm explaining such a partial pressure dependence is the Temkin-isotherm

$$q = \zeta \ln \left( \frac{K_T p}{\zeta} \right),\tag{5}$$

see the supplement S1. Inserting the derivative of Eq. (5) in Eq. (1) yields the prevailing pressure dependence in the limit $p \to 0$

$$5 \quad k' = \frac{2RT}{r} \frac{\zeta}{p}.\tag{6}$$

For the transition between the high and low concentration limits we assume

$$k' = \frac{2RT}{r} \left( \frac{\zeta}{p} + K_H \right).\tag{7}$$

to be valid. Note that only the model parameter $\zeta$ of the Temkin isotherm can be determined from concentration dependent $k'$-measurements. This is because the second model parameter $K_T$ is arbitrarily removed by differentiation of Eq. (5). The

10 linear relationship of $k'$ vs. $1/p$ is used to study the relationship between film thickness, temperature, pre-humidification and column aging which are treated as independent variables by multiple linear regression using data from 336 chromatograms.





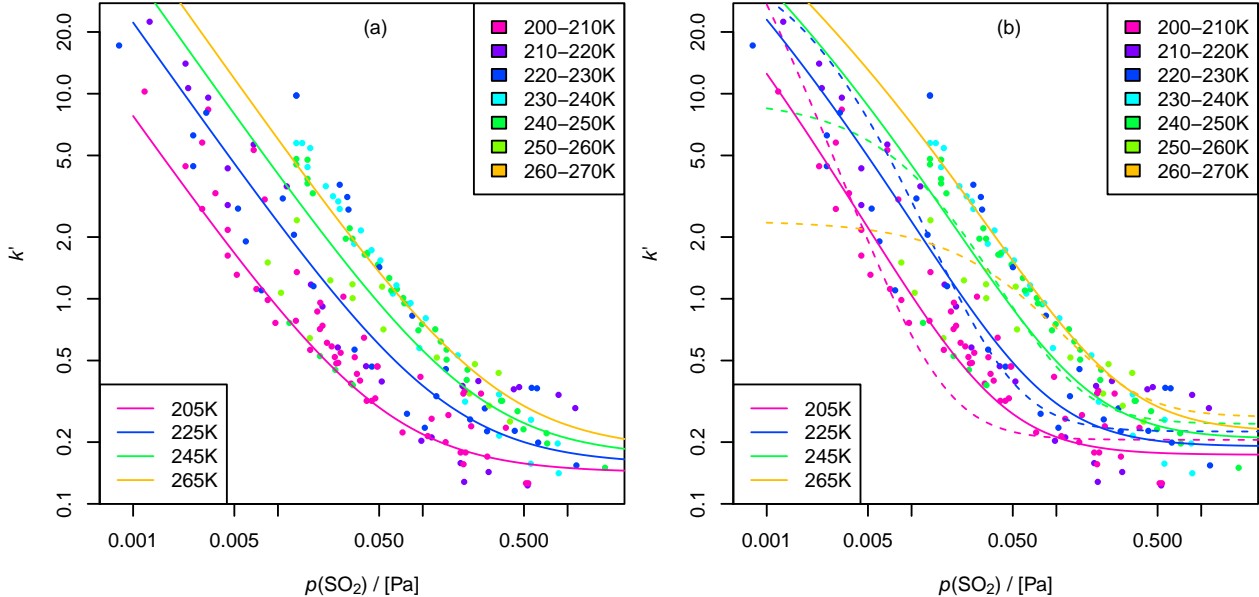

**Figure 3.** Double logarithmic plot of $k'$ vs. $p(SO_2)$ at peak maximum for experiments with ice film thickness $h > 4$ $\mu$m and without ice aging. (a) Fit with Temkin adsorption isotherm according to Eq. (7). (b) Fit with simple Langmuir model (dashed lines) according to Eq. (10) and with the dissociative Langmuir model (solid curves) according to Eq. (16).

Since the retention times at lower concentrations are difficult to obtain due to considerable peak broadening, the obtained $k'$ are weighted with $1/k'$ in the linear regression. The results indicate that $k'$ significantly increases with increasing temperature ($P < 0.01$), column aging ($P < 0.01$) and ice film thickness ($P = 0.03$). Therefore, further analysis is confined to a subset of 249 experiments with ice film thickness $h > 4$ $\mu$m. All these experiments were performed using pre-humidified carrier

gas. In this subset, $k'$ significantly increases with temperature ($P < 0.01$) and column aging ($P < 0.01$) but not with ice film thickness ($P = 0.81$). A linear dependency of $\zeta$ from $T$ and column aging is formulated, see Table 1. Next, the fit parameters are determined by nonlinear regression of the log-scaled data, see Fig. 3a. The residuals of the fit are normal distributed (Shapiro-Wilk normality test, $P = 0.19$).

Langenberg (1997) applied a Langmuir adsorption model for the chemisorption in addition to a Henry's adsorption isotherm

for weak physisorption. The Langmuir-isotherm is defined by the expression

$$\theta = \frac{q}{q_S} = \frac{(K_L/q_S)p}{1 + (K_L/q_S)p},\tag{8}$$

where $q_S$ is the monolayer saturation capacity. For $K_L$, a van't Hoff like temperature dependency was assumed. For low $p$ where $(K_L/q_S)p \ll 1$ the Langmuir-model simplifies to the Henry's adsorption isotherm $q \approx K_L p$. By applying Eq. (1) to the Langmuir-model of Eq. (8), we form

$$k' = \frac{2RT}{r} \frac{K_L q_S^2}{(K_L p + q_S)^2}\tag{9}$$



and by adding Henry's law adsorption isotherm arithmetic expression for independent weak physisorption one obtains

$$k' = \frac{2RT}{r} \left( \frac{K_L q_S^2}{(K_L p + q_S)^2} + K_H \right). \tag{10}$$

When fitting this model to the log-scaled data, the dashed lines in Fig. 3b are obtained with the model parameters listed in Table 1. From these model parameters, an apparent adsorption enthalpy of $\Delta H_a = -(41 \pm 3)\ \mathrm{kJ\,mol^{-1}}$ is estimated. As can be seen in Fig. 2b, the fit is quite good at lower temperatures but fails at higher temperatures. When comparing the fit to those of the Temkin model, it is obvious that the Langmuir-model is not appropriate.

Clegg and Abbatt (2001) proposed a model for the adsorption of $SO_2$ where $SO_2$ interacts with the ice surface by hydrolyzing, as it does when being dissolved in liquid water. The equilibrium of the first dissociation step of $SO_2$ in water (see R1) is described by

$$K_{I1} = \frac{q(H_3O^+) q(HSO_3^-)}{q(SO_2)} \tag{11}$$

On neutral ice, where $q(HSO_3^-) = q(H_3O^+)$, the surface concentration of $S_{IV}$ is given by

$$q(S_{IV}) = q(HSO_3^-) + q(SO_2) \tag{12}$$
$$= \sqrt{K_{I1} K_H p} + K_H p. \tag{13}$$

Together with Eq. (1), we arrive at

$$k' = \frac{2RT}{r} \left( \frac{1}{2} \sqrt{\frac{K_H K_{I1}}{p}} + K_H \right). \tag{14}$$

This model yields a dependence of $k' \propto p^{-1/2}$ in the low concentration regime, but not for our data which were obtained at much higher concentrations than those used by Clegg and Abbatt (2001).

In cases where an adsorbing molecule dissociates upon adsorption the Langmuir isotherm takes a modified form (Crowley et al., 2010; Huthwelker et al., 2006):

$$\theta = \frac{q}{q_S} = \frac{\sqrt{\frac{K_H K_{I1} p}{q_S^2}}}{1 + \sqrt{\frac{K_H K_{I1} p}{q_S^2}}}. \tag{15}$$

With Eq. (1) and adding the Henry-adsorption term, it follows that

$$k' = \frac{2RT}{r} \left( \frac{\frac{K_H K_{I1}}{q_S}}{2\sqrt{\frac{K_H K_{I1} p}{q_S^2}} \left( \sqrt{\frac{K_H K_{I1} p}{q_S^2}} + 1 \right)^2} + K_H \right). \tag{16}$$

Model parameters $K_H K_{I1}$ and $q_S$ are determined by nonlinear regression. No temperature dependency of $K_H K_{I1}$ is found. The dependency $q_S$ from temperature is formulated by a van't Hoff like expression, see Table 1. In addition, $q_S$ is about 60% larger for aged ice. However, the residuals of the fit are not normal distributed (Shapiro-Wilk normality test, $P = 0.04$).





Due to strong scatter, we are not able to find a significant temperature trend for $K_H$ describing the physisorption of $SO_2$. To explore the temperature dependency of $K_H$, temperature dependent measurements of $k'$ were performed using higher concentrations with a partial pressure of around $p(SO_2) = 1$ Pa at the recorded peak maximum. In addition, the experimental setup was changed: instead of using the methylsilyl-deactivated column, an untreated fused silica column was used. We as-

sumed that due to better wettability, ice deposition on the untreated silica surface results in a better surface coverage than in methylsilyl-deactivated columns. Furthermore, the coating procedure in these subsets of experiments was modified to minimize the length of the uncoated part of the column inside the cold box: after preparing the coating as described above, the cold box was opened and the uncoated tail of the column was manually drawn out of the box though a pinhole. This reduced the length of the uncoated tail in the cold box to about 30 cm. For further stabilization of the ice surface, the column was maintained

for 3 h at 256 K after the coating procedure. After keeping the column at 207 K overnight, the chromatographic experiments were performed the next day. The ice film thickness was 6.5 $\mu$m. To find evidence for anomalous adsorption behavior of $SO_2$ caused by the formation of a quasi-liquid layer when approaching the melting point of water ice, the $SF_6/SO_2$ chromatograms where measured up to 267 K. If a surface premelt layer acted like a supercooled liquid water film, $SO_2$ might dissolve in the layer therefore enhancing the capacity ratio $k'$ and therefore $K_H$ when approaching the melting point. However, with one

exception where the column erroneously had not been rinsed before with Milli-Q water and dried with synthetic air to remove soluble impurities, mainly $NO_3^-$-ions, $SF_6$ and $SO_2$ peaks could not be separated at temperatures $> 232$ K. This implies that our results obtained with the untreated column are unaffected by a quasi-liquid layer at higher $p(SO_2)$ where $k'$ is nearly independent of $p(SO_2)$. The results indicate that the dominant adsorption at higher $p(SO_2)$ which was probed in our experiments is physisorption on the dry surface. In those cases where the measured peaks of $SF_6$ and $SO_2$ overlap the true peak maxima

were determined by fitting two exponential modified Gauss functions to the overlapping peaks (Felinger, 1994). Afterwards the peak maximum time of each peak was determined by a linear search algorithm. $K_H$ at $p \to \infty$ is determined as intercept by linear regression of $k'$ vs. $p^{-1}$ using Eq. (7). In Fig. 4 the Henry's law adsorption constant $\ln K_H$ obtained for ice coated methylsilyl-deactivated columns and an ice coated untreated column are plotted against $1/T$. With van't Hoff equation

$$\ln K_H = \ln K_H^\infty - \frac{\Delta H_a}{RT} \tag{17}$$

no reasonable fit is possible for the experiments performed with methylsilyl-deactivated columns. Even in the case of the untreated column, the values of $K_H$ systematically deviate from the van't Hoff plot regression line. Whereas $K_H$ is weakly dependent of temperature using the methylsilyl-deactivated fused silica column, $K_H$ decreases more strongly with temperature than when using an untreated fused silica column. A possible explanation for this might be that at higher temperatures formation of a quasi-liquid layer occurs at the interface of ice with the methylsilyl-deactivated fused silica surface.

**3.2 Adsorption of acetone and *n*-hexane**

To explore the possible impact of a quasi-liquid layer, the adsorption of acetone and *n*-hexane on an ice coated methylsilyl-deactivated column was probed in the temperature range 211 - 265 K. These compounds were selected because they are comparable in vapor pressure over a wide temperature range but differ in water solubility: *n*-hexane is insoluble in water,



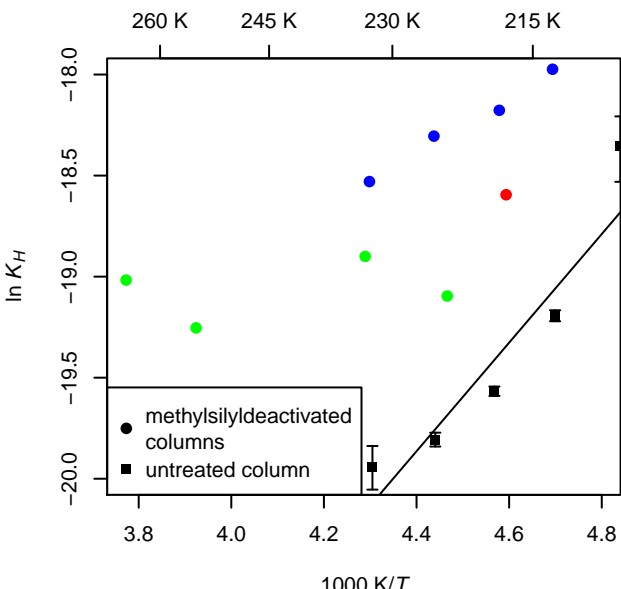

**Figure 4.** $K_H$ as function of $T$ for experiments using an untreated column and methylsilyl-deactivated columns coated with aged ice. Experiments conducted with the same prepared column are marked by the same color.

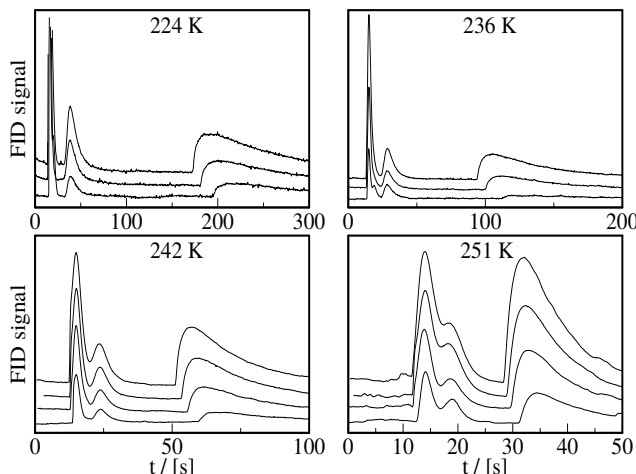

**Figure 5.** Chromatograms of a mixture of methane, *n*-hexane and acetone at different temperatures. Flow rate 10 sccm. Methane is used as non adsorbable tracer.

whereas acetone is mixable with water. Methane was used as inert tracer. Fig. 5 shows the chromatograms obtained using a methylsilyl-deactivated column coated with an ice film of 7.1 $\mu$m thickness. From left to right, the recorded peaks are





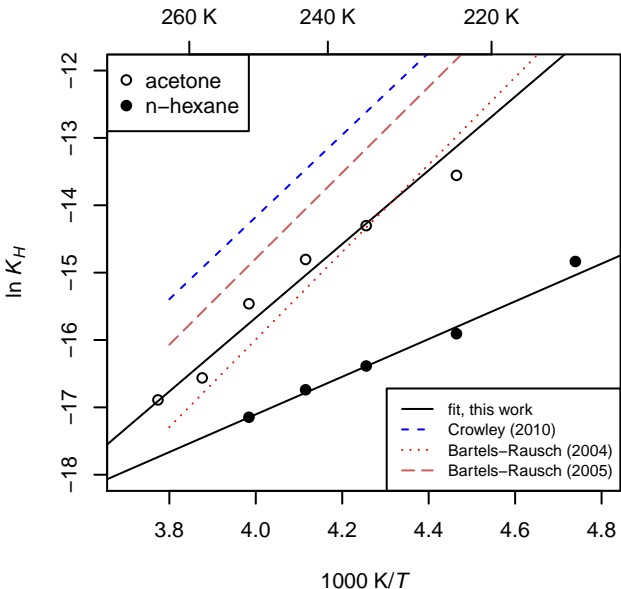

**Figure 6.** Van't Hoff plot of $\ln K_H$ vs. inverse temperature for adsorption of acetone and $n$-hexane on ice using a methylsilyl-deactivated fused silica column. The solid lines are fits of the data to Eq. (17), used to derive adsorption enthalpies. Van't Hoff plots of acetone from Bartels-Rausch et al. (2004a, b) and Bartels-Rausch et al. (2005) and recommended values from Crowley et al. (2010) extrapolated from lower temperatures to the temperature range investigated are shown for comparison.

from methane, $n$-hexane, and acetone. The $n$-hexane peaks are nearly symmetrical and the retention times are independent of the amounts injected, whereas the acetone peaks show some tailing with slightly increasing retention times with decreasing injection amounts, indicating that the slope of the isotherm is nonlinear. The adsorption of acetone on ice is described by a Langmuir adsorption isotherm according to Eq. (8), see review of Crowley et al. (2010). Eq. (9) is reorganized into a linear

5 relation of $1/\sqrt{k'}$ vs. $p$:

$$\sqrt{\frac{2RT}{rk'}} = \frac{\sqrt{K_L}}{q_S}p + \frac{1}{\sqrt{K_L}}. \tag{18}$$

$K_L$ of acetone is determined after linear regression from the intercept. For $n$-hexane, mean values of the retention times are determined for different injection amounts at one temperature. $K_H$ of $n$-hexane is calculated using Eq. (4). The enthalpies of adsorption are determined using Eq. (17) by linear regression, see Fig. 6 and Table 2.





**Table 2.** Adsorption isotherm of *n*-hexane and acetone on ice

|  | $\ln K_H^\infty / 1\,\mathrm{mol\,m^{-2}\,Pa^{-1}}$ | $\Delta H_a$ [kJ mol$^{-1}$] |
| --- | --- | --- |
| *n*-hexane | $-28.3 \pm 0.9$ | $-23 \pm 2$ |
| acetone | $-37.6 \pm 2.5$ | $-45 \pm 5$ |

## 4 Discussion

### 4.1 Consistency of the peak maxima method

The consistency of the peak maxima method is checked by theoretical peak shape calculations, see supplement S2 and Fig. 2b. When comparing locations of experimental peak maxima with peak maxima predicted by Eq. (1), it can be seen that at higher concentrations, the peak maxima are shifted towards longer retention times. As stated by Jönsson and Lövkvist (1987), for Langmuir-like adsorption isotherms the accuracy of the peak maxima method is best for low to moderate concentrations. Hence, in case of nonlinear adsorption, the isotherm determined by the peak maxima method at higher concentration has to be regarded as upper limit.

Slow uptake of $SO_2$ by diffusion into the ice surface is included into the 2D model, see Fig. 2b. When taking diffusion into the ice surface into account, the peak maxima are shifted on the curve of Eq. (1) to higher retention times. In addition, the peak maxima are smaller and tailing is enhanced. As can bee seen from the simulation without including slow uptake (1D model), Eq. (1) is nearly the envelope of the tails of the simulated peaks. This is used by the *peak profile method* (Huber and Gerritse, 1971) to determine the slope of the isotherm from a single peak tail. But, due to slow uptake into the surface, this method is not applicable for $SO_2$ on ice.

### 4.2 Comparison with literature

#### 4.2.1 Adsorption of $SO_2$

The reanalysis of our experimental data reveals that the simple Langmuir adsorption isotherm is not suitable for the description of adsorption of $SO_2$ on ice. Only considering our data, either the Temkin model or the dissociative Langmuir model are more appropriate. When taking the results of Clegg and Abbatt (2001) and the chemical nature of $SO_2$ into account, we conclude that the adsorption is best described by the dissociative Langmuir model of Eq. (15) combined with simple physisorption described by Henry's adsorption isotherm. The experiments of Clegg and Abbatt (2001) were performed at lower $SO_2$ concentrations, thus saturation effects were not observed. $SO_2$ reaction with ice and dissociation is also supported by the study of Jagoda-Cwiklik et al. (2008) who investigated $SO_2$ adsorbates on the surface of ice nanoparticles at 128 K by FTIR spectroscopy: like in aqueous solutions, an $HSO_3^-$ ion was found.





Crowley et al. (2010) reported a value of $K_{I1}K_H = 1.3 \times 10^{-13}$ mol$^2$ m$^{-4}$ Pa$^{-1}$ based on the data of Clegg and Abbatt (2001) for the model of Eq. (13) at 228 K. This value lies within the error limit of our value of $K_{I1}K_H = (7 \pm 5) \times 10^{-14}$ mol$^2$ m$^{-4}$ Pa$^{-1}$. A limitation of our study is that the experiments were performed at rather high SO$_2$ concentrations. This makes the determination of $K_{I1}K_H$ rather difficult, as can bee seen by a value of $P = 0.1$ for this fit parameter, see

Table 1. Therefore, we are not able to find a temperature trend for $K_{I1}K_H$. This is in contrast to the observation of Clegg and Abbatt (2001) who found less uptake at lower temperatures.

The monolayer saturation capacity $q_S$ of the dissociative Langmuir model describes the maximal adsorbed amount of SO$_2$ in one monolayer. Our values range from $9 \times 10^{-9}$ mol m$^{-2} \triangleq 5 \times 10^{11}$ cm$^{-2}$ at 205 K to $4 \times 10^{-8}$ mol m$^{-2} \triangleq 2 \times 10^{12}$ cm$^{-2}$ at 265 K, see Table 1. Clegg and Abbatt (2001) estimated that about $5 \times 10^{14}$ cm$^{-2}$ molecules of the size of SO$_2$ can be

packed on the surface next to each other. For small molecules a saturation capacity of $2.2 \times 10^{14}$ - $4.5 \times 10^{14}$ cm$^{-2}$ was found experimentally (Crowley et al., 2010). Taking $5 \times 10^{14}$ cm$^{-2}$ as reference, dissociative adsorption of SO$_2$ must occur on active sites representing 0.1% - 0.5% of the total surface. This result may be explained by surface premelting. At temperatures close to the melting point the ice surface may be regarded as a quasi liquid layer where the hexagonal oxygen lattice is completely distorted. At temperatures below $\approx 260$ K the oxygen lattice is distorted by point defects and the hydrogen-bond

network is distorted. Either vacancies in the outer bilayer predicted by molecular dynamics simulations (Bishop et al., 2009; Riikonen et al., 2014) or Hydronium, hydroxide, and the Bjerrum L- and D-defects predicted by density functional theory calculations (Watkins et al., 2010) prevailing at the surface come into question as active sites for dissociative adsorption. D-defects at the surface, bearing a positive charge, possibly could attract HSO$_3^-$-ions at the surface. Watkins et al. (2010) found by density functional calculation that surface D-defects form with a very small energy penalty of 0.06 eV. From model parameter

$a_2$ describing the temperature dependency of $q_S$, an activation energy of 11 kJ mol$^{-1} \triangleq 0.11$ eV for the formation of an active site can be derived. Surprisingly for the hypothetical case of $T \to \infty$ K the saturation capacity approaches a limit of $q_s = 3 \times 10^{14}$ cm$^{-2}$.

One unanticipated finding is that $q_S$ is larger for aged ice. A possible explanation for this might be that surface defects are more prevalent on the surface of aged ice. Another possible explanation for this is that the ice surface in our capillary columns

increases during aging: shortly after preparation of the column, the inner wall is covered with individually frozen drops of water. During aging, the interstitial ice-free surface of the column is eventually covered with ice by desublimation from the frozen drops.

Besides the aging effect, other reasons for poor reproducibility of the properties of ice surfaces must exists. Whereas the reproducibility of the chromatograms of one column at one temperature is good, the reproducibility of experiments of two

prepared columns is rather bad. There are several possible explanations for this result: the properties of the ice surface is also affected by the underlying surface properties of the fused silica column, which may influence cristallinity and surface defects.

### 4.2.2  Adsorption of acetone and *n*-hexane

In recent years, several studies concerning the adsorption of acetone on ice have been published, see the review of Crowley et al. (2010) and references therein. However, all these studies only covered the temperature range of $140 - 228$ K. In this review, the





results are summarized as an expression for the van't Hoff equation for Henry's law adsorption isotherm using Eq. (17). When extrapolating these expressions to the higher temperature range of our experiment, it is found that the isotherms of Bartels-Rausch et al. (2004a, b) and Bartels-Rausch et al. (2005) are closest to our measured values of Henry's adsorption isotherm constant, see Fig. 6. In the first study, breakthrough curves were measured using a column packed with ice beads and in the second study an ice coated-wall flow-tube was used. Bartels-Rausch et al. (2004a) and Bartels-Rausch et al. (2005) obtained an adsorption enthalpy of $-52 \pm 2\,\mathrm{kJ\,mol^{-1}}$ and $-46 \pm 3\,\mathrm{kJ\,mol^{-1}}$, respectively. Our value of $-45 \pm 5\,\mathrm{kJ\,mol^{-1}}$ is slightly lower but complies within the error limits.

Bartels-Rausch et al. (2004a) determined the surface area available for adsorption by a BET analysis of methane adsorption isotherms. Bartels-Rausch et al. (2005) prepared the ice surface by slowly freezing water at the inner surface of a Pyrex flow-tube. They assume that the ice surface corresponds to the geometric inner surface of the tube. Thus, in both experiments the surface of ice adsorbents is regarded to be known. These results suggest that within experimental uncertainty, the real ice surface in our column corresponds approximately to the geometric inner surface of the ice coated part of the column, due to the consistency of our data concerning the adsorption of acetone with these studies. The temperature trend of the isotherms of Bartels-Rausch et al. (2004a) and Bartels-Rausch et al. (2005) extrapolates to higher temperatures up to 265 K. No visible change caused by a quasi-liquid layer is found in the temperature trend.

For $n$-hexane on pure ice, the enthalpy of adsorption has not been measured yet. Hoff et al. (1995) reported $\Delta H_a = -37.3 \pm 1.3\,\mathrm{kJ\,mol^{-1}}$ for the adsorption on ice-coated Chromosorb P in the temperature range $263 - 273$ K, which is much higher than our value of $-23 \pm 2\,\mathrm{kJ\,mol^{-1}}$. For adsorption of $n$-hexane on liquid water $-28\,\mathrm{kJ\,mol^{-1}}$ were determined by Hartkopf and Karger (1973) at 286 K.

## 4.3 Atmospheric implications

The adsorbed fraction of a trace gas in a contrail or cirrus cloud with ice surface area density $\sigma$ is given by

$$\frac{N_s}{N_g} = \frac{\sigma R T q}{p}. \tag{19}$$

In case of Henry's adsorption isotherm of Eq. (3), we can write

$$\frac{N_s}{N_g} = \sigma R T K_H. \tag{20}$$

In case of the model Eq. (13) and of the dissociative Langmuir model of Eq. (15), we arrive at

$$\frac{N_s}{N_g} \approx \sigma R T \sqrt{\frac{K_{I1} K_H}{p}} \tag{21}$$

at low concentrations. For cirrus clouds $\sigma = 0.003\,\mathrm{m^{-1}}$ and for contrails $\sigma = 0.003 - 0.012\,\mathrm{m^{-1}}$ were reported by Schröder et al. (2000). In an aircraft plume $SO_2$-concentrations of $> 5 \times 10^8\,\mathrm{cm^{-3}}$ were observed by Schumann et al. (1998). At 205 K the adsorbed fraction calculated with Eq. (21) is about $10^{-3}$ for $\sigma = 0.003\,\mathrm{m^{-1}}$. For acetone the adsorbed fraction according to Eq. (20) is about $10^{-4}$ at 205 K. These findings suggest that adsorption of $SO_2$ and acetone on ice particles in the plume of an aircraft and subsequent vertical transport by particle sedimentation is negligible.





Sokolov and Abbatt (2002) estimated an upper limit for the surface density of $\sigma = 0.1 \, \text{m}^{-1}$ for tropospheric ice clouds. Here, the adsorbed fraction of $SO_2$ is about 0.3 at the same $SO_2$ concentration as above. Therefore, we conclude that interaction of $SO_2$ with ice is only important in very dense ice clouds.

## 5   Conclusions

Gas-chromatography with water-ice coated fused silica column is a well suited screening tool to study adsorption of trace gases with weak or medium adsorption in the range $k' = 0.1 - 100 \triangleq K_H = 8 \times 10^{-9} - 8 \times 10^{-6} \, \text{mol m}^{-2}\text{Pa}^{-1}$. By using humidified carrier gas, the experiments can be performed at temperatures close below the melting point. Excepting the special low temperature coating box, only standard equipment for gas chromatography is required.

The interaction of $SO_2$ with ice surfaces in the temperature range $205 - 265$ K is described by a dissociative Langmuir-model

in conjunction with weaker physisorption described by Henry's adsorption isotherm. No temperature trend for the adsorption equilibrium constant is found for the dissociative adsorption process. The monolayer saturation capacity for dissociative adsorption increases with increasing temperature and ice aging. Adsorption occurs on active sites representing 0.1% - 0.5% of the total surface. These findings support the existence of surface premelting.

Slow uptake into the ice surface is evidenced by peak tailing in addition to tailing arising from the nonlinear isotherm.

The possible interference of the underlying bare or methylsilyl-deactivated column surface with the ice surface cannot be ruled out. Further work is required to find evidence if ice films on untreated fused silica surfaces or on methylsilyl-deactivated fused silica surfaces are more likely resembling the surface of atmospheric ice particles.

*Code availability.*   The R-code for simulation of chromatograms is provided as supplement.

*Author contributions.*   U.S. suggested the experimental setup. S.L. performed the experiments, analyzed the data, and drafted the manuscript.

*Competing interests.*   The authors declare that they have no conflict of interest.

*Acknowledgements.*   This work was supported by the "Deutsche Forschungsgemeinschaft (DFG)" within the DFG-priority program "Basics of the Impact of Air and Space Transportation on the Atmosphere". We thank Thomas Huthwelker for helpful discussions.





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
