# Peer review of "Gas-Chromatography using ice coated fused silica-columns: Study of adsorption of sulfur dioxide on water-ice"

_Atmospheric Chemistry and Physics, 2017_

## Referee Comment (RC1) · Anonymous Referee #1 · 3 Jan 2018

This is an excellent piece of scientific work that fills a gap owing to the chosen temperature range of 205 to 265 K, a range which most studies do not explore. In addition, it uses a simple experimental technique that is rarely used in the present context despite its widespread use in separation sciences. I highly appreciate the present work as it corresponds to a thorough and thoughtful experimental study including random error and uncertainty estimates as well as the confrontation of experimental data with three models of SO2 surface interaction at increasing complexity. It always appeals to compare different methods of experimental techniques for a given problem of some importance in order to gauge the possible existence of measurement artifacts or fallacies as a confidence building principle. The present study undertakes exactly this which

contributes to its importance and value.

At this point I would like to raise a few questions that the authors should try to answer during the peer-review process in order to help the reader grasp the full significance of the work:

- The slow and inexorable "Degradation" of the ice film owing to the carrier gas decompression (pg. 3, lines 19 and 20, and equation (1)) should be explained in a bit more detail. Is this a temperature effect owing to adiabatic decompression across the length of the capillary? If one has a flow with an equilibrium amount of humidity in the carrier gas one loses this same amount at the exit to first order: What goes in must come out at the end of the capillary. What you lose by sublimation is redeposited downstream, provided we are at steady-state and have a uniform temperature profile. What else is implied or important beyond that? Any unaccounted loss processes? Please explain the scientific basis or complications.

- I strongly suggest the addition of a qualitative explanation for the concentration dependence of the SO2 retention time displayed in Figure 2. As far as I understand this effect it comes from the (partial) saturation of the SO2 uptake (at equilibrium): The higher the SO2 concentration the earlier the breakthrough because of vanishing interaction with the ice owing to surface saturation of the uptake following equation (11) (Langmuir dissociation, strong interaction) and (3) (Henry adsorption isotherm, weak interaction).

- The Temkin isotherm (equation (5)) seems to be a parametric treatment of the above behavior according to the derivation in the S1 section: Is there any scientifically rooted explanation to simply add algebraically both contributions (Henry + each of the stronger interactions) in all three models displayed in Table 1 as an "interpolation" or superposition scheme of two extremes? The molecular view (saturation to a variable degree depending on the SO2 partial pressure) naturally arrives at the same result for purely kinetic reasons. Diffusion tube experiments under molecular flow conditions (T. Koch
et al., JPCA 1998; C. Alcala-Jornod et al., PCCP 2000; C. Alcala-Jornod et al., JPCA 2004) arrived at the identical saturation behavior of salt and ice interfaces (without the presence of a carrier gas).

- What is the rationale for using "deactivated" (alkyl-silylated) Pyrex as a Substrate to grow the ice sample in the first place? I do not believe that a coating of several micrometers will let the SO2 "feel" the underlying properties of the substrate. From our own studies of pure ice performing quartz crystal microbalance measurements on the evaporation rate of H2O from pure ice films result in a value of 80 to 100 nm thick layer beyond which subsurface effects cease to be important. Beyond this thickness the kinetics of evaporation is unchanged up to several micrometer thickness of pure vapor-deposited ice which is believed to be less compact (lower density, more surface imperfections) than ice samples frozen from pure water. By the same token, an interference of the silylated glass interface with ice at several micrometers thickness is highly unlikely (pg. 13, lines 30 and 31, and pg. 15, lines 15 and 16) unless the authors have solid evidence to the contrary.

- Regarding the strongly tailing peak shapes of the chromatograms for SO2, and to a lesser extent for acetone: Is this a thermodynamic or kinetic effect? Is the equilibrium between adsorbed and gas phase SO2 established at low concentrations of SO2 at constant flow rate?

- As a general remark the units of KH, KL and q should be clearly included, at least once when mentioned in the text for the first time. One has to be aware that Henry's adsorption isotherm (KH) is different from Henry's law solubility constant (gas-bulk partition coefficient)!

In addition, there are several minor technical points that may be raised:

- Regarding the square root dependence of the surface coverage (equation (15) and pg. 8, line 16) the requirement is that the active sites must be NEIGHBORING sites in order to yield the square root dependence. - Pg. 2, line 30 and 31: sentence does

not make sense! - Pg. 3, line 20: "...ice film slowly but unavoidably..." - Pg. 4, lines 15 and 16: Incomplete sentence. - Pg. 8, line 5: I do not see a temperature effect in Figure 2b. It probably must be Figure 3b. - Pg. 7, Figure 3: I do not see yellow data points, however there is a yellow (fitted) line. - Pg. 10, bottom: "miscible". - Pg. 12, Table 2: units of KH: mol m-2Pa-1.

Please also note the supplement to this comment:
https://www.atmos-chem-phys-discuss.net/acp-2017-800/acp-2017-800-RC1-supplement.pdf

---

## Referee Comment (RC2) · Anonymous Referee #3 · 21 Feb 2018

The study presents results from solid-gas chromatography of several trace gases to ice. The work covers a T range of 205 to 265 K. From these, conclusions about the partitioning of SO2 to ice in the upper troposphere and of the role of surface disorder on the partitioning are drawn. The partitioning of SO2 to ice has raised quite a discussion in the community and is far from being understood. Further, the role of surface disorder is a key-topic. This makes the topic of the manuscript highly relevant for ACP.

I'm not convinced that this manuscript presents new and innovative results and fear the technical content is addressing only a very small and specific part of the atmospheric science community. I also find it difficult to capture the results based on the

data presented. While I acknowledge the re-analysis and am enthusiastic about process modelling to derive fundamental data from chromatographic results, I'm sorry to suggest rejecting this manuscript.

* The gas-phase concentrations of SO2 seem too high to me. Extrapolation from experiments at such high concentrations to environmental concentrations – what are typical concentrations of SO2 in air masses in contact with ice or snow anyway, please specify in introduction – is highly questionable for a number or reasons as shown for a number of trace gases (see later work on HNO3 by Abbatt group). This lets me question the environmental relevance of this work. * The high concentrations obviously results in very high formal surface coverages, so that SO2-SO2 interactions can not be excluded. I don't understand the use of the Henry or Langmuir parameterization in this context – which strictly speaking works best at low coverage. What is the surface coverage at the peak position in your columns? * I don't understand why your surface saturation capacity is so low? To me this looks like there is something odd with the analysis. Could you convince me with the acetone data that your approach is working? * May I ask you to stick to the IUPAC nomenclature. So, your Henry would become KLinC, for example. * Working with SO2 and in acknowledgement of Huthwelkers work highlighting the role of solvation into liquid pockets, I strongly suggest to discuss the phase diagram. Taken that the freezing point depression by SO2 is rather modest, I do not expect a large impact but clarification is needed.

Last, the manuscript remains very technical without a clear discussion on what is to be learned from fitting the chromatographic results.

---

## Author Response (AR1)

This is an excellent piece of scientific work that fills a gap owing to the chosen temperature range of 205 to 265 K, a range which most studies do not explore. In addition, it uses a simple experimental technique that is rarely used in the present context despite its widespread use in separation sciences. I highly appreciate the present work as it corresponds to a thorough and thoughtful experimental study including random error and uncertainty estimates as well as the confrontation of experimental data with three models of SO2 surface interaction at increasing complexity. It always appeals to compare different methods of experimental techniques for a given problem of some importance in order to gauge the possible existence of measurement artifacts or fallacies as a confidence building principle. The present study undertakes exactly this which

contributes to its importance and value.

At this point I would like to raise a few questions that the authors should try to answer during the peer-review process in order to help the reader grasp the full significance of the work:

- The slow and inexorable "Degradation" of the ice film owing to the carrier gas decompression (pg. 3, lines 19 and 20, and equation (1)) should be explained in a bit more detail. Is this a temperature effect owing to adiabatic decompression across the length of the capillary? If one has a flow with an equilibrium amount of humidity in the carrier gas one loses this same amount at the exit to first order: What goes in must come out at the end of the capillary. What you lose by sublimation is redeposited downstream, provided we are at steady-state and have a uniform temperature profile. What else is implied or important beyond that? Any unaccounted loss processes? Please explain the scientific basis or complications.

- I strongly suggest the addition of a qualitative explanation for the concentration dependence of the SO2 retention time displayed in Figure 2. As far as I understand this effect it comes from the (partial) saturation of the SO2 uptake (at equilibrium): The higher the SO2 concentration the earlier the breakthrough because of vanishing interaction with the ice owing to surface saturation of the uptake following equation (11) (Langmuir dissociation, strong interaction) and (3) (Henry adsorption isotherm, weak interaction).

- The Temkin isotherm (equation (5)) seems to be a parametric treatment of the above behavior according to the derivation in the S1 section: Is there any scientifically rooted explanation to simply add algebraically both contributions (Henry + each of the stronger interactions) in all three models displayed in Table 1 as an "interpolation" or superposition scheme of two extremes? The molecular view (saturation to a variable degree depending on the SO2 partial pressure) naturally arrives at the same result for purely kinetic reasons. Diffusion tube experiments under molecular flow conditions (T. Koch

et al., JPCA 1998; C. Alcala-Jornod et al., PCCP 2000; C. Alcala-Jornod et al., JPCA 2004) arrived at the identical saturation behavior of salt and ice interfaces (without the presence of a carrier gas).

- What is the rationale for using "deactivated" (alkyl-silylated) Pyrex as a Substrate to grow the ice sample in the first place? I do not believe that a coating of several micrometers will let the SO2 "feel" the underlying properties of the substrate. From our own studies of pure ice performing quartz crystal microbalance measurements on the evaporation rate of H2O from pure ice films result in a value of 80 to 100 nm thick layer beyond which subsurface effects cease to be important. Beyond this thickness the kinetics of evaporation is unchanged up to several micrometer thickness of pure vapor-deposited ice which is believed to be less compact (lower density, more surface imperfections) than ice samples frozen from pure water. By the same token, an interference of the silylated glass interface with ice at several micrometers thickness is highly unlikely (pg. 13, lines 30 and 31, and pg. 15, lines 15 and 16) unless the authors have solid evidence to the contrary.

- Regarding the strongly tailing peak shapes of the chromatograms for SO2, and to a lesser extent for acetone: Is this a thermodynamic or kinetic effect? Is the equilibrium between adsorbed and gas phase SO2 established at low concentrations of SO2 at constant flow rate?

- As a general remark the units of KH, KL and q should be clearly included, at least once when mentioned in the text for the first time. One has to be aware that Henry's adsorption isotherm (KH) is different from Henry's law solubility constant (gas-bulk partition coefficient)!

In addition, there are several minor technical points that may be raised:

- Regarding the square root dependence of the surface coverage (equation (15) and pg. 8, line 16) the requirement is that the active sites must be NEIGHBORING sites in order to yield the square root dependence. - Pg. 2, line 30 and 31: sentence does

not make sense! - Pg. 3, line 20: "...ice film slowly but unavoidably..." - Pg. 4, lines 15 and 16: Incomplete sentence. - Pg. 8, line 5: I do not see a temperature effect in Figure 2b. It probably must be Figure 3b. - Pg. 7, Figure 3: I do not see yellow data points, however there is a yellow (fitted) line. - Pg. 10, bottom: "miscible". - Pg. 12, Table 2: units of KH: mol m-2Pa-1.

Please also note the supplement to this comment:
https://www.atmos-chem-phys-discuss.net/acp-2017-800/acp-2017-800-RC1-supplement.pdf
* * *
Atmos. Chem. Phys. Discuss.,
https://doi.org/10.5194/acp-2017-800-AC1, 2018

[Figure]

We thank the referee for reviewing and commenting our discussion paper. The remarks of the reviewer are marked like *this*. To the comments and questions we answer as follows:

**Reply to specific comments and questions**

*- The slow and inexorable "Degradation" of the ice film owing to the carrier gas decompression (pg. 3, lines 19 and 20, and equation (1)) should be explained in a bit more detail. Is this a temperature effect owing to adiabatic decompression across the length*

*of the capillary? If one has a flow with an equilibrium amount of humidity in the carrier gas one loses this same amount at the exit to first order: What goes must come out at the end of the capillary. What you lose by sublimation is redeposited downstream, provided we are at steady-state and have a uniform temperature profile. What else is implied or important beyond that? Any unaccounted loss processes? Please explain the scientific basis or complications.*

The flow through the column is described by Poiseuille's equation for compressible fluids. It is assumed that the temperature of the carrier is permitted to equilibrate with the column wall. Due to the decompression of the carrier gas along the column at constant temperature, the mixing water vapor in equilibrium with the ice surface increases along the flow tube. Therefore, while the mass flow rate of the carrier gas $\dot{n}$ is constant, the mass flow rate of water vapor increases along the ice-coated column.

It is assumed that the water vapor pressure $p(H_2O)$ is in equilibrium with the ice surface. The mass flow of water leaving the column is given by

$$\dot{n}_o(H_2O) = \frac{p(H_2O)\dot{V}_o}{RT}$$

where $\dot{V}_o$ is the volume flow rate of the carrier gas at column exit. It is calculated from the carrier gas mass flow rate $\dot{n}$ and the carrier gas pressure at column exit $p_o$:

$$\dot{V}_o = \frac{\dot{n}RT}{p_o}.$$

Since the carrier gas entering the column is pre-humidified, the water mass flow entering the column is given by

$$\dot{n}_i(H_2O) = \frac{p(H_2O)\dot{V}_i}{RT}.$$

Due to carrier gas decompression $\dot{V}_i < \dot{V}_o$. Therefore, less water is entering the column

than leaving. Hence, the net rate of water leaving the column is given by

$$\dot{n}(H_2O) = \dot{n}_o(H_2O) - \dot{n}_i(H_2O) = \dot{n}p(H_2O)\left(\frac{1}{p_o} - \frac{1}{p_i}\right).$$

After calculating $\dot{h}$ from $\dot{n}(H_2O)$, Eq. (2) is obtained.

We conclude that our adsorption experiments were performed under desublimation conditions with respect to $H_2O$.

*- I strongly suggest the addition of a qualitative explanation for the concentration dependence of the $SO_2$ retention time displayed in Figure 2. As far as I understand this effect it comes from the (partial) saturation of the $SO_2$ uptake (at equilibrium): The higher the $SO_2$ concentration the earlier the breakthrough because of vanishing interaction with the ice owing to surface saturation of the uptake following equation (11) (Langmuir dissociation, strong interaction) and (3) (Henry isotherm, weak interaction).*

An explanation has been added to the introduction (p3, l5).

*- The Temkin isotherm (equation (5)) seems to be a parametric treatment of the above behavior according to the derivation in the S1 section: Is there any scientifically rooted explanation to simply add algebraically both contributions (Henry + each of the stronger interactions) in all three models displayed in Table 1 as an "interpolation" or superposition scheme of two extremes? The molecular view (saturation to a variable degree depending on the $SO_2$ partial pressure) naturally arrives at the same result for purely kinetic reasons. Diffusion tube experiments under molecular flow conditions (T. Koch et al., JPCA 1998; C. Alcala-Jornod et al., PCCP 2000; C. Alcala-Jornod et al., JPCA 2004) arrived at the identical saturation behavior of salt and ice interfaces (without the presence of a carrier gas).*

As we describe, the modified Temkin and Langmuir isotherms pass over to Henry's law of adsorption at very low concentration. But this type of adsorption must be a stronger chemisorption like adsorption process in contrast to the weaker physisorp-

tion process. Both adsorption processes occur in parallel over the whole concentration range. Therefore, at very low concentration, the effective Henry's adsorption isotherm constant would be $K_H + K_T$. Even in case of dissociative adsorption, a linear adsorption behavior would be expected when $q(H_3O^+) > q(HSO_3^-)$.

*- What is the rationale for using "deactivated" (alkyl-silylated) Pyrex as a Substrate to grow the ice sample in the first place? I do not believe that a coating of several micrometers will let the $SO_2$ "feel" the underlying properties of the substrate. From our own studies of pure ice performing quartz crystal microbalance measurements on the evaporation rate of $H_2O$ from pure ice films result in a value of 80 to 100 nm thick layer beyond which subsurface effects cease to be important. Beyond this thickness the kinetics of evaporation is unchanged up to several micrometer thickness of pure vapor-deposited ice which is believed to be less compact (lower density, more surface imperfections) than ice samples frozen from pure water. By the same token, an interference of the silylated glass interface with ice at several micrometers thickness is highly unlikely (pg. 13, lines 30 and 31, and pg. 15, lines 15 and 16) unless the authors have solid evidence to the contrary.*

Due to experimental constraints, about 1 m of the column inside the box remained uncoated. In order to minimize interfering adsorption of $SO_2$ on the ice-free surface, we used a methylsilyl-deactivated column. This is explained in the revised manuscript (p4, l18). We agree that $SO_2$ adsorption on the ice coating is not likely to be affected by the properties of the underlying tube surface. It is more likely that certain ice film properties e.g. surface roughness depend on the substrate. Unfortunately the microphysical properties of the ice coatings in our capillary columns could not be probed by our experimental techniques.

*- Regarding the strongly tailing peak shapes of the chromatograms for $SO_2$, and to a lesser extent for acetone: Is this a thermodynamic or kinetic effect? Is the equilibrium between adsorbed and gas phase $SO_2$ established at low concentrations of $SO_2$ at constant flow rate?*

The most likely cause of the tailing is nonlinear adsorption at low temperature. This is supported by the decreasing adjusted retention times tn with increasing amounts of acetone injected. Therefore, Eq. (18) was used to extrapolate our measurements to zero concentration. However, we cannot exclude additional tailing due to slow incorporation of acetone into the ice surface even at higher temperatures. We assume that adsorption equilibrium of $SO_2$ on ice is established with respect to physisorption and dissociative adsorption but not with respect to uptake into the ice surface under our experimental conditions.

*- As a general remark the units of $K_H$ , $K_L$ and $q$ should be clearly included, at least once when mentioned in the text for the first time. One has to be aware that Henry's adsorption isotherm ($K_H$) is different from Henry's law solubility constant (gas-bulk partition coefficient)!*

$K_H$, $K_T$ and $K_L$ have the same unit. This makes them comparable between themselves. After first occurrence of the symbols in the text, the unit has been denoted now in the revised manuscript.

The term *Henry's law of adsorption* was adopted from Wikipedia, see https://en. wikipedia.org/wiki/Henry_adsorption_constant: "The Henry adsorption constant is the constant appearing in the linear adsorption isotherm, which formally resembles Henry's law; therefore, it is also called Henry's adsorption isotherm."

*- Regarding the square root dependence of the surface coverage (equation (15) and pg. 8, line 16) the requirement is that the active sites must be NEIGHBORING sites in order to yield the square root dependence.*

This is required for the steady state derivation of Eq. (15) given by Huthwelker et. al (2006). However, Eq. (15) is a thermodynamic relation. It can also be derived as follows on a pure thermodynamic base:

We assume that $HSO_3^-$ is trapped by an unknown ice surface defect X. In addition to

equilibrium (R1) there is a second equilibrium

$$HSO_3^- + X \rightleftharpoons XHSO_3^-.$$

The total surface concentration of X is limited to $q_S$. This equilibrium corresponds to Langmuir adsorption of $HSO_3^-$ on ice.

**Reply to proposed technical corrections**

*- Pg. 2, line 30 and 31: sentence does not make sense!*

Fixed.

*- Pg. 3, line 20: "...ice film slowly but unavoidably..."*

Fixed.

*- Pg. 4, lines 15 and 16: Incomplete sentence.*

Fixed.

*- Pg. 8, line 5: I do not see a temperature effect in Figure 2b. It probably must be Figure 3b.*

Fixed.

*- Pg. 7, Figure 3: I do not see yellow data points, however there is a yellow (fitted) line.*

Due to a strange software error, some points were not colored correctly. A workaround has been applied. A color palette better suited for printout has been chosen.

*- Pg. 10, bottom: "miscible".*

Fixed.

*- Pg. 12, Table 2: units of $K_H$: $mol\, m^{-2}\, Pa^{-1}$.*

$\ln K_H$ must be dimensionless, therefore $K_H$ must be divided by $1\ \mathrm{mol\,m^{-2}\,Pa^{-1}}$. To clarify this, two braces have been added.

The revised marked-up version of the discussion paper is attached as supplement.

Please also note the supplement to this comment:
https://www.atmos-chem-phys-discuss.net/acp-2017-800/acp-2017-800-AC1-supplement.pdf

———————————————————

Atmos. Chem. Phys. Discuss.,
https://doi.org/10.5194/acp-2017-800-RC2, 2018

[Figure]
The study presents results from solid-gas chromatography of several trace gases to ice. The work covers a T range of 205 to 265 K. From these, conclusions about the partitioning of SO2 to ice in the upper troposphere and of the role of surface disorder on the partitioning are drawn. The partitioning of SO2 to ice has raised quite a discussion in the community and is far from being understood. Further, the role of surface disorder is a key-topic. This makes the topic of the manuscript highly relevant for ACP.

I'm not convinced that this manuscript presents new and innovative results and fear the technical content is addressing only a very small and specific part of the atmospheric science community. I also find it difficult to capture the results based on the

data presented. While I acknowledge the re-analysis and am enthusiastic about process modelling to derive fundamental data from chromatographic results, I'm sorry to suggest rejecting this manuscript.

* The gas-phase concentrations of SO2 seem too high to me. Extrapolation from experiments at such high concentrations to environmental concentrations – what are typical concentrations of SO2 in air masses in contact with ice or snow anyway, please specify in introduction – is highly questionable for a number or reasons as shown for a number of trace gases (see later work on HNO3 by Abbatt group). This lets me question the environmental relevance of this work. * The high concentrations obviously results in very high formal surface coverages, so that SO2-SO2 interactions can not be excluded. I don't understand the use of the Henry or Langmuir parameterization in this context – which strictly speaking works best at low coverage. What is the surface coverage at the peak position in your columns? * I don't understand why your surface saturation capacity is so low? To me this looks like there is something odd with the analysis. Could you convince me with the acetone data that your approach is working? * May I ask you to stick to the IUPAC nomenclature. So, your Henry would become KLinC, for example. * Working with SO2 and in acknowledgement of Huthwelkers work highlighting the role of solvation into liquid pockets, I strongly suggest to discuss the phase diagram. Taken that the freezing point depression by SO2 is rather modest, I do not expect a large impact but clarification is needed.

Last, the manuscript remains very technical without a clear discussion on what is to be learned from fitting the chromatographic results.

Atmos. Chem. Phys. Discuss.,
https://doi.org/10.5194/acp-2017-800-AC2, 2018

[Figure]

We thank the referee for reviewing and commenting our discussion paper. The remarks of the reviewer are marked like *this*. All references, symbols and equations used and cited herein refer to the discussion paper unless otherwise indicated. To the comments and questions we answer as follows:

**Reply to general comments**

We consider the manuscript worth to be published in ACP:

- A new gas chromatographic technique is presented to study trace gas – ice interactions.

- A dissociative adsorption mechanism of $SO_2$ on ice limited to distinguished active sites is proposed. The mechanism should be validated in future by theoretical calculations.

**Reply to specific comments and questions**

*- The gas-phase concentrations of $SO_2$ seem too high to me. Extrapolation from experiments at such high concentrations to environmental concentrations – what are typical concentrations of $SO_2$ in air masses in contact with ice or snow anyway, please specify in introduction – is highly questionable for a number of reasons as shown for a number of trace gases (see later work on $HNO_3$ by Abbatt group). This lets me question the environmental relevance of this work.*

We agree that the high concentrations are a limitation of our study. Just our experiments show that trace gas ice interactions dramatically can change when changing the concentration range.

In the lower troposphere $SO_2$ concentrations $< 50$ ppbv (Heikes et al., 1987, http://dx.doi.org/10.1029/JD092iD01p00915) are found. This corresponds to $p(SO_2) < 0.005$ Pa which is at the lower concentration range of our study.

Our study was motivated by the assumption of vertical trace gas transport in the wake of an aircraft by adsorption and subsequent sedimentation. Higher $SO_2$ concentrations than in the troposphere were expected in the plume. However, later Schumann et al. (1998) showed that the plume is rapidly diluted with ambient air. Therefore, within few seconds, the $SO_2$ concentration drops to $5 \times 10^8 - 2 \times 10^{10}$ cm$^{-3}$ which is about two decimal powers lower than the concentration range investigated. However, we could show that our measurements are consistent with the work of Clegg and Abbatt (2001) when extrapolating our model to lower concentrations.

*- The high concentrations obviously results in very high formal surface coverages, so that SO$_2$-SO$_2$ interactions can not be excluded. I don't understand the use of the Henry or Langmuir parameterization in this context – which strictly speaking works best at low coverage. What is the surface coverage at the peak position in your columns?*

Weak physisorption and dissociative adsorption at active sites are different processes and must be considered separately.

Regarding the physisorption of SO$_2$ at the normal ice surface: using Eq. (8) and $K_L = 1.35 \times 10^{-8}$ mol m$^{-2}$Pa$^{-1}$, only a surface coverage $\theta = 0.2\%$ is found at a peak partial pressure of $p(\text{SO}_2) = 1$ Pa. Thus, SO$_2$-SO$_2$ interactions can be neglected.

Regarding adsorption at the active sites: using Eq. (15) at 205 K the degree of saturation of the active sites is $\theta = 97\%$ and at 266 K $\theta = 87\%$.

Therefore, during the chromatographic experiments nearly complete saturation of the active sites but not of normal surface is achieved at peak maximum.

*- I don't understand why your surface saturation capacity is so low? To me this looks like there is something odd with the analysis. Could you convince me with the acetone data that your approach is working?*

The existence of active sites responsible for dissociative adsorption is also surprising for us. The surface saturation capacity for dissociative adsorption is so low, because dissociative adsorption must be caused only by active sites, see discussion section in our paper.

For acetone, we did not find any evidence for adsorption at active sites. However, this is not surprising because acetone does not dissociate in aqueous solution. Tailing observed for the acetone peaks is caused by saturation of the normal ice surface.

Regarding the adsorption of acetone at the normal ice surface, our data are consistent with other studies published previously, see Fig. 6.

*- May I ask you to stick to the IUPAC nomenclature. So, your Henry would become $K_{LinC}$, for example.*

We assume that the referee refers to Crowley et al. (2010), a report of the IUPAC Subcommittee on Gas Kinetic Data Evaluation for Atmospheric Chemistry. Therein, the linear adsorption isotherm is defined as

$$K_{\text{linC}} = [\text{X}]_s/[\text{X}]_g.$$

with $[K_{\text{linC}}] = $ cm. $K_{\text{linC}}$ and $K_H$ can easily be converted by

$$K_{\text{linC}} = RTK_H.$$

Also therein, the Langmuir isotherm is defined as

$$\theta = \frac{K_{\text{LangC}}[\text{X}]_g}{1 + K_{\text{LangC}}[\text{X}]_g}$$

with $[K_{\text{LangC}}] = $ cm$^3$. $K_{\text{LangC}}$ is related to $K_L$ by

$$K_{\text{LangC}} = \frac{RTK_L}{q_s}.$$

The units are different and therefore, $K_{\text{linC}}$ is not comparable with $K_{\text{LangC}}$. In the IUPAC report cm is used as base unit. This may facilitate the integration of the data into kinetic atmospheric models. Outside this application, using cm as base unit is error-prone, since unit conversions are required. Therefore, we formulated the adsorption isotherms purely with SI-units as function of partial pressure instead of particle density, what we do not intend to change. This has the advantage that adsorption enthalpies directly can be obtained from the van't Hoff plot.

*- Working with SO$_2$ and in acknowledgment of Huthwelkers work highlighting the role of solvation into liquid pockets, I strongly suggest to discuss the phase diagram. Taken*

*that the freezing point depression by SO$_2$ is rather modest, I do not expect a large impact but clarification is needed.*

This subject was already discussed by Huthwelker et al. (2001): Rather large SO$_2$ partial pressures higher than in our experiments are required to form aqueous SO$_2$ solutions or the SO$_2$·6H$_2$O hydrate. They also report the phase diagram of the system H$_2$O-SO$_2$.

To achieve melting of the ice surface by freezing point depression, a solute must be solved in the ice surface with a mole fraction

$$x_s > \frac{\Delta H_m \Delta T}{RT_m^2}$$

where $\Delta H_m = 6008$ J mol$^{-1}$ is the melting enthalpy of ice and $T_m$ the melting temperature.

The partial pressure of the trace gas must exceed a certain threshold value to achieve melting of the ice surface. $x_s(\text{SO}_2)$ and $p(\text{SO}_2)$ are interconnected by Henry's law yielding

$$p(\text{SO}_2) \approx \frac{x_s(\text{SO}_2)}{M(\text{H}_2\text{O})H}$$

where $M(\text{H}_2\text{O}) = 0.018$ kg mol$^{-1}$ and $H$ is the Henry coefficient of SO$_2$ in supercooled water. The coefficient is obtained from data of water after extrapolating to lower temperatures, see Langenberg et al. (1998). Therefore, for the temperature limits of our experiment, the following values are obtained, assuming full dissociation of SO$_2$:

| $T$ [K] | $\Delta T$ [K] | $x_s(\text{SO}_2)$ | $H$ [mol kg bar$^{-1}$] | $p(\text{SO}_2)$ [Pa] |
|---|---|---|---|---|
| 205 | 68 | 0.66 | 797 | $> 2300$ |
| 265 | 8 | 0.08 | 4 | $> 6 \times 10^4$ |

Since the partial pressure of SO$_2$ in our experiment is much lower, surface melting by freezing point depression is not expected. However, the freezing point of ice may be depressed in veins and nodes. Huthwelker et al. (2001) postulated that slow uptake of SO$_2$ is caused by uptake in these liquid reservoirs. We also observed slow uptake, but we did not analyze it quantitatively.

The revised manuscript was not changed anymore, it is attached as supplement in final form with all changes applied.

Please also note the supplement to this comment:
https://www.atmos-chem-phys-discuss.net/acp-2017-800/acp-2017-800-AC2-supplement.pdf

[revised manuscript text omitted]

---

## Author Response (AR2)

**Revised Submission: Reply to anonymous referee #3**

Stefan Langenberg[1,*] and Ulrich Schurath[2]

[1]Institut für Physikalische und Theoretische Chemie, University of Bonn, Germany
[*]now at: Klinik und Poliklinik für Hals-Nasen-Ohrenheilkunde/Chirurgie, University of Bonn, Germany
[2]Institut für Umweltphysik, University of Heidelberg, Germany

**Correspondence:** Stefan Langenberg (langenberg@uni-bonn.de)

We thank referee #3 for reviewing and commenting our manuscript again and reconsidering for publication. The remarks of the reviewer are marked like *this*. All references, symbols and equations used and cited herein refer to the discussion paper unless otherwise indicated. All references to pages (p) and lines (l) of our reply refer to the revised marked-up manuscript in this file. To the comments and questions we answer as follows:

*Thank you for your detailed answer and comments that have clarified some of my concern. I'd still like to ask for a major revision because I feel that some key characteristics of the experiments are not discussed detailed enough.*

**1  *Surface coverage**

*It is of paramount importance to state surface coverages during the experiments early in the manuscript.*

- *The reader needs to be assured that the experiments probed the $SO_2$-ice interaction and were not done in the multilayer regime of the adsorption isotherms where $SO_2$-$SO_2$ interaction would have been probed.*

- *High surface coverage can lead to an overload of chromatographic column which results in tailing. By addressing surface coverage in the columns during the experiments early in the manuscript, this artifact needs to be ruled out.*

- *Surface coverage is also a key parameter to compare different experiments and characterize the experiments. If one wants to compare the findings of this study to other flow tube studies of uptake of acidic gases to ice (such as Crowley's work on HCl) one need to make sure that similar settings prevailed, most important similar surface coverages.*

*I hope that the surface coverage can be estimated without using the adsorption isotherms to parameterize the results. It would be most useful to derive the coverage as directly form the chromatographs as possible also giving the opportunity to compare this estimate to the results of the Langmuir-Henry parameterization. As the amount and concentration of injected sample is*

*known, estimates of surface coverage could maybe be gained based on the resolving power of the column to estimate the surface area of ice at which the interaction equilibrium is established as the peak moves through the column OR based on the partition coefficient as given in Eq. 1:*

$$k' = t_n/t_0 = \frac{Conc(\text{stationary phase})}{Conc(\text{mobile phase})} * \frac{vol(\text{stationary phase})}{vol(\text{mobile phase})}.$$

*Your answer to my previous comment makes me confident that neither of these artifacts are an issue, but I'd like you to confirm and describe this in more details based on the raw data (or as close to the raw data as possible). Taken together this should address both the tailing — to rule out column overload — and the shift of the chromatographic peaks with time - to ensure that the weaker interaction (shorter retention time) is not caused by $SO_2$ condensation or multilayer adsorption but by $SO_2$-ice interaction.*

We included a discussion of the saturation in the results section of the revised manuscript, see p6, l8 – p7, l2.

**2    *Temperature trend**

*An „anomalous" temperature trend of $SO_2$ sorption to ice has been described in earlier studies: While most trace gases show increased sorption with decreasing temperature, $SO_2$ uptake increased with increasing temperature. Such behavior seems to be typical for acidic trace gases and has been explained by a combination of the temperature trend of adsorption and that of the acid-base dissociation (Zimmermann et al., 2016). However, we are far from solving this puzzle and adding to this would make this manuscript indeed highly interesting for ACP community. I therefore ask to go into details. Could you show a Figure similar to Figure 2 of chromatograms at constant surface coverage but decreasing temperature? Or, you could base the discussion on Figure 3: At 0.05 Pa $SO_2$ k increases from 210-220 = 220-230 K < 200-210K < 260-270 K < 230-240 = 240-250 = 250-26K. I'm a little concerned about the scatter, that you openly address throughout the manuscript. Restricting this discussion to a subset might be a good idea, or a detailed discussion of surface coverage would probably need to be considered. Any uptake model needs to be able to explain this particular temperature trend. It might help the reader to state and discuss the temperature trend in detail early in the manuscript. (On page 7 line 9, you state that a subset of experiments shows a T dep of k'. This statement is put in the context of discussing the Temkin isotherm. As k' is an observable, I'd suggest to put this discussion upfront and establish your observed T trend first).*

It is important to differentiate between adsorption and uptake. We investigated the adsorption of $SO_2$ on ice. During the experiments we also observed uptake of $SO_2$, but we could not analyze it quantitatively.

Fig. 2 was obtained using one specific coated column at one specific temperature. For a comparison of chromatograms at a given $SO_2$ partial pressure over a wide range of temperatures (as you suggested e.g. for 0.05 Pa), it would be necessary to select chromatograms with similar partial pressures at peak maximum, since the retention times are sensitive functions of the

temperature. However, the actual concentrations at the detected peak maxima could not be controlled during our experiments. We screened our chromatograms for peaks with accidentally similar concentrations at the maxima, but recorded at different temperatures, as suggested. However, we could not find an appropriate subset of chromatograms.

We added a hint to the $k'$ - temperature dependency in the results section early, p7, l5 of the revised manuscript. However, the temperature dependency of $k'$ can only be derived from the linear regression.

The dependency of $\zeta$ of the Temkin-model can be displayed as follows: For $k' > 1$, the Henry's adsorption isotherm can be neglected and Eq. (6) is valid. Eq. (6) is rearranged to

$$\zeta = \frac{k' r p(\text{SO}_2)}{2RT}.$$

$\zeta$ is plotted versus the temperature for a subset of 77 experiments with $k' > 1$:

[Figure]

The linear regression line reveals a significant ($P < 0.01$) temperature dependence of the Temkin parameter $\zeta$.

**3   *Relevance of Henry vs. Temkin/Langmuir**

*Taken that two uptake-mechanism working hand-in-hand describe your data well, the question arises on the relative importance of either. If I understood correctly, the Temkin isotherm or the dissociative Langmuir explain the change in DHads with surface coverage at low concentrations and the Henry explains the increase in peak area at short residence time (weaker*

*adsorption) at high concentrations. Could you elaborate on the relative importance of either isotherm when going from low to high temperatures and from low to high concentrations. At which Conc and T does the Henry kick in? In particular, taken that the adsorption sites for dissociation are so very low, the amount of $SO_2$ that adsorbed dissociatively might be negligible compared to the total amount? Could you prove me wrong by discussion the fraction of $SO_2$ the adsorption of which flows Henry and which follows dis-langmuir.*

We cast your question referring to the importance of the dissociative Langmuir adsorption relative to the total amount of adsorbed $SO_2$ by defining the ratio

$$y = \frac{q(\text{diss. Langmuir})}{q(\text{Henry}) + q(\text{diss. Langmuir})}.$$

Using Eq. (4) and Eq. (16) and the extracted model parameters listed in Table 1, the ratio $y$ can be calculated as function of $p(SO_2)$ and temperature:

[Figure]

As can be seen, at low partial pressures most of the $SO_2$ is dissociatively adsorbed at a limited number of active sites. The fraction of $SO_2$ that is additionally adsorbed on the normal ice surface increases linearly and dominates at high partial pressures of $SO_2$ where the active sites are saturated. This is now briefly addressed in the discussion on p15, 120 of the revised manuscript.

*Thank you very much for considering this and I hope you find the comments helpful.*

**4  *Minor comments:**

*please add a description of Langmuir and Temkin isotherms somewhere early in the text.What are the key basics of these concepts.*

Added to the revised manuscript, see p7, l8 and p8, l14.

*p3 line 15: how is the thickness of the ice determined?*

Added, see p3, l15–19.

*p5 line 23: What was the concentration of $SO_2$, methane, hexane, acetone that was dosed to the column?*

Added, see p4, l28–32.

*p 10 ff: Please state concentration and surface coverage of these species as well.*

Added, see p6, l10–12 and p12, l1–4.

*p10 line 10: I'm a little concerned about the tailing and shift that you see with acetone. If the acid-base equilibrium is causing this for $SO_2$, this should be absent in acetone data. Could this rather be an effect of surface coverage?*

Yes, of course, see p12, l4–6.

**References**

Zimmermann, S., Kippenberger, M., Schuster, G., and Crowley, J. N.: Adsorption isotherms for hydrogen chloride (HCl) on 
[revised manuscript text omitted]